# When are Local Queries Useful for Robust Learning?

**Pascale Gourdeau**
University of Oxford
pascale.gourdeau@cs.ox.ac.uk

**Varun Kanade**
University of Oxford
varunk@cs.ox.ac.uk

**Marta Kwiatkowska**
University of Oxford
marta.kwiatkowska@cs.ox.ac.uk

**James Worrell**
University of Oxford
james.worrell@cs.ox.ac.uk

## Abstract

Distributional assumptions have been shown to be necessary for the robust learnability of concept classes when considering the exact-in-the-ball robust risk and access to random examples by Gourdeau et al. (2019). In this paper, we study learning models where the learner is given more power through the use of *local* queries, and give the first *distribution-free* algorithms that perform robust empirical risk minimization (ERM) for this notion of robustness. The first learning model we consider uses local membership queries (LMQ), where the learner can query the label of points near the training sample. We show that, under the uniform distribution, LMQs do not increase the robustness threshold of conjunctions and any superclass, e.g., decision lists and halfspaces. Faced with this negative result, we introduce the local *equivalence* query (LEQ) oracle, which returns whether the hypothesis and target concept agree in the perturbation region around a point in the training sample, as well as a counterexample if it exists. We show a separation result: on one hand, if the query radius $\lambda$ is strictly smaller than the adversary's perturbation budget $\rho$, then distribution-free robust learning is impossible for a wide variety of concept classes; on the other hand, the setting $\lambda = \rho$ allows us to develop robust ERM algorithms. We then bound the query complexity of these algorithms based on online learning guarantees and further improve these bounds for the special case of conjunctions. We finish by giving robust learning algorithms for halfspaces with margins on both $\{0, 1\}^n$ and $\mathbb{R}^n$.

## 1 Introduction

Adversarial examples have been widely studied since the work of (Dalvi et al., 2004; Lowd and Meek, 2005a,b), and later (Biggio et al., 2013; Szegedy et al., 2013), the latter having coined the term. As presented in Biggio and Roli (2017), two main settings exist for adversarial machine learning: *evasion* attacks, where an adversary perturbs data at test time, and *poisoning* attacks, where the data is modified at training time.

The majority of the guarantees and impossibility results for evasion attacks are based on the existence of adversarial examples, potentially crafted by an all-powerful adversary. However, what is considered to be an adversarial example has been defined in two different, and in some respects contradictory, ways in the literature. The *exact-in-the-ball* notion of robustness (also known as *error region* risk in Diochnos et al. (2018)) requires that the hypothesis and the ground truth agree in the perturbation region around each test point; the ground truth must thus be specified on all input points in the perturbation region. On the other hand, the constant-in-the-ball notion of robustness (which is also known as *corrupted input* robustness from the work of Feige et al. (2015)) requires that the unperturbed point be correctly classified and that the points in the perturbation region share its label,

meaning that we only need access to the test point labels; see, e.g., (Diochnos et al., 2018; Dreossi et al., 2019; Gourdeau et al., 2021; Pydi and Jog, 2021) for thorough discussions on the subject.

We study the problem of robust classification against evasion attacks under the exact-in-the-ball definition of robustness. Previous work for this problem, e.g., (Diochnos et al., 2020; Gourdeau et al., 2021), has considered the setting where the learner only has access to random examples. However, many defences against evasion attacks have used adversarial training, the practice by which a dataset is augmented with previously misclassified points. Moreover, in the learning theory literature, some learning models give more power to the learner, e.g., by using membership and equivalence queries. Our work studies the robust learning problem mentioned above from a learning theory point of view, and investigates the power of *local queries* in this setting.

## 1.1 Our Contributions

We outline our contributions below. All our results use the *exact-in-the-ball* definition of robustness. Conceptually, we study the powers and limitations of robust learning with access to oracles that only reveal information nearby the training sample. Our results are particularly relevant as they contrast with the impossibility of robust learning in the *distribution-free* setting when only random examples are given, as demonstrated in Gourdeau et al. (2019).

**Limitations of the Local Membership Query Model.** In the local membership query (LMQ) model, the learner is allowed to query the label of points in the vicinity of the training sample. This model was introduced by Awasthi et al. (2013) and shown to guarantee the PAC learnability of various concept classes (which are believed or known to be hard to learn with only random examples) under distributional assumptions. However, we show that LMQs do not improve the robustness threshold of the class of conjunctions under the uniform distribution. Indeed, any $\rho$-robust learning algorithm will need a joint sample and query complexity that is exponential in $\rho$, and thus superpolynomial in the input dimension $n$ against an adversary that can flip $\rho = \omega(\log n)$ input bits at test time.

**The Local Equivalence Query Model.** Faced with the query lower bound for LMQ above, one may consider giving a different power to the learner to improve robust learning guarantees. We thus introduce the local equivalence query (LEQ) model, where the learner is allowed to query whether a hypothesis and the ground truth agree in the vicinity of points in the training sample. The LEQ oracle is the natural exact-in-the-ball analogue of the Perfect Attack Oracle introduced in Montasser et al. (2021), which was developed for the constant-in-the-ball robustness. It is also a variant of the equivalence query oracle introduced by Angluin (1987).

**Distribution-Free Robust ERM with an LEQ Oracle.** We show that having access to a *robustly consistent* learner (i.e., one that can get zero robust risk on the training sample) gives sample complexity upper bounds that are logarithmic in the size of the hypothesis class or linear in its *robust* VC dimension–a complexity measure adapted from Cullina et al. (2018) for our notion of robustness, which we present in this paper. We study the setting where the learner has access to random examples and an LEQ oracle. In the case where the query radius $\lambda$ of the LEQ oracle is strictly smaller than the adversarial perturbation budget $\rho$, we show that, for a wide variety of concept classes, distribution-free robust learning is impossible, regardless of the training sample size. In contrast, when $\lambda = \rho$ we exhibit robustly consistent learners that use an LEQ oracle. This separation result further validates the need for an LEQ oracle in the distribution-free setting. We furthermore use online learning setting results to exhibit upper bounds on the LEQ oracle query complexity and then improve these bounds in the specific case of conjunctions. Finally, we study the sample and query complexity of halfspaces on both $\{0, 1\}^n$ and $\mathbb{R}^n$. To our knowledge, the results presented in this paper feature the first robust empirical risk minimization (ERM) algorithms for the *exact-in-the-ball* robust risk in the literature.[1]

## 1.2 Related Work

**Learning with Membership and Equivalence Queries.** Membership and equivalence queries (MQ and EQ, respectively) have been widely used in learning theory. Membership queries allow the

---

[1]Note that previous work, e.g., Gourdeau et al. (2021), used PAC learning algorithms as black boxes, which are not in general robust risk minimizers, unless they also happen to be exact learning algorithms, and that (Montasser et al., 2019; 2021) use the constant-in-the-ball definition of robustness.

learner to query the label of any point in the input space $\mathcal{X}$, namely, if the target concept is $c$, MQ returns $c(x)$ when queried with $x \in \mathcal{X}$. The goal is usually to learn the target $c$ exactly. Recall that, in the probabilistically approximately correct (PAC) learning model of Valiant (1984), the learner has access to the example oracle $\mathsf{EX}(c, D)$, which upon being queried returns a point $x \sim D$ sampled from the underlying distribution and its label $c(x)$, and the goal is to output $h$ such that with high probability $h$ has low error.[2] The EQ oracle takes as input a hypothesis $h$ and returns whether $h = c$, and provides a counterexample $z$ such that $h(z) \neq c(z)$ otherwise. The seminal work of Angluin (1987) showed that deterministic finite automata (DFA) are exactly learnable with a polynomial number of queries to MQ and EQ in the size of the DFA. Many classes were then showed to be learnable in this setting as well as others, see e.g., (Bshouty, 1993; Angluin, 1988; Jackson, 1997). Moreover, the MQ + EQ model has recently been used for recurrent and binarized neural networks (Weiss et al., 2018, 2019; Okudono et al., 2020; Shih et al., 2019), and interpretability (Camacho and McIlraith, 2019). But even these powerful learning models have limitations: learning DFAs only with EQ is hard (Angluin, 1990) and, under cryptographic assumptions, they are also hard to learn solely with the MQ oracle (Angluin and Kharitonov, 1995). It is also worth noting that the MQ learning model has been criticized by the applied machine learning community, as labels can be queried in the whole input space, irrespective of the distribution that generates the data. In particular, (Baum and Lang, 1992) observed that query points generated by a learning algorithm on the handwritten characters oftentimes appeared meaningless to human labellers. Awasthi et al. (2013) thus offered an alternative learning model to Valiant's original model, the PAC and local membership query (EX + LMQ) model, where the learning algorithm is only allowed to query the label of points that are close to examples from the training sample. Bary-Weisberg et al. (2020) later showed that many concept classes, including DFAs, remain hard to learn in the EX + LMQ model.

**Existence of Adversarial Examples.** It has been shown that, in many instances, the vulnerability of learning models to adversarial examples is inevitable due to the nature of the learning problem. The majority of the results have been shown for the constant-in-the-ball notion of robustness, see e.g., (Fawzi et al., 2016, 2018a,b; Gilmer et al., 2018; Shafahi et al., 2018; Tsipras et al., 2019). As for the exact-in-the-ball definition of robustness, Diochnos et al. (2018) consider the robustness of monotone conjunctions under the uniform distribution. Using the isoperimetric inequality for the boolean hypercube, they show that an adversary that can perturb $O(\sqrt{n})$ bits can increase the misclassification error from 0.01 to $1/2$. Mahloujifar et al. (2019) then generalize this result to Normal Lévy families and a class of well-behaved classification problems (i.e., ones where the error regions are measurable and average distances exist).

**Sample Complexity of Robust Learning.** Our work uses a similar approach to Cullina et al. (2018), who define the notion of adversarial VC dimension to derive sample complexity upper bounds for robust ERM algorithms, with respect to the constant-in-the-ball robust risk. Montasser et al. (2019) use the same notion of robustness and show sample complexity upper bounds for robust ERM algorithms that are polynomial in the VC and dual VC dimensions of concept classes, giving general upper bounds that are exponential in the VC dimension–though they sometimes must be achieved by an improper learner. Ashtiani et al. (2020) build on their work and delineate when proper robust learning is possible. On the other hand, (Khim et al., 2019; Yin et al., 2019; Awasthi et al., 2020) study *adversarial* Rademacher complexity bounds for robust learning, giving results for linear classifiers and neural networks when the robust risk can be minimized (in practice, this is approximated with adversarial training). Viallard et al. (2021) derive PAC-Bayesian generalization bounds for the averaged risk on the perturbations, rather than working in a worst-case scenario. As for the exact-in-the-ball definition of robustness, Diochnos et al. (2020) show that, for a wide family of concept classes, any learning algorithm that is robust against all $\rho = o(n)$ attacks must have a sample complexity that is at least an exponential in the input dimension $n$. They also show a superpolynomial lower bound in case $\rho = \Theta(\sqrt{n})$. Gourdeau et al. (2019) show that distribution-free robust learning is generally impossible. They also show that monotone conjunctions have a robustness threshold of $\Theta(\log n)$ under log-Lipschitz distributions, meaning that this class is efficiently robustly learnable against an adversary that can perturb $\log n$ bits of the input, but if an adversary is allowed to perturb $\rho = \omega(\log n)$ bits of the input, there does not exist a sample-efficient learning algorithm for this problem. Gourdeau et al. (2021) extended this result to the class of monotone decision lists

---

[2]This is known as the realizable setting. It is also possible to have a distribution over the labels, in which case we are working in the *agnostic* setting.

and Gourdeau et al. (2022) showed a sample complexity lower bound for monotone conjunctions that is exponential in $\rho$ and that the robustness threshold of decision lists is also $\Theta(\log n)$. Finally, Diakonikolas et al. (2020) and Bhattacharjee et al. (2021) have used online learning algorithms for robust learning with respect to the constant-in-the-ball notion of robustness.

**Restricting the Power of the Learner and the Adversary.** Most adversarial learning guarantees and impossibility results in the literature have focused on all-powerful adversaries. Recent work has studied learning problems where the adversary's power is curtailed. E.g, Mahloujifar and Mahmoody (2019) and Garg et al. (2020) study the robustness of classifiers to polynomial-time attacks. Closest to our work, Montasser et al. (2020, 2021) study the sample and query complexity of robust learning with respect to the constant-in-the-ball robust risk when the learner has access to a Perfect Attack Oracle (PAO). For a perturbation type $\mathcal{U} : \mathcal{X} \to 2^{\mathcal{X}}$, hypothesis $h$ and labelled point $(x, y)$, the PAO returns the constant-in-the-ball robust loss of $h$ in the perturbation region $\mathcal{U}(x)$ and a counterexample $z$ where $h(z) \neq y$ if it exists. Our LEQ oracle is the natural analogue of the PAO oracle for our notion of robustness. In the constant-in-the-ball *realizable* setting,[3] the authors use online learning results to show sample and query complexity bounds that are linear and quadratic in the Littlestone dimension of concept classes, respectively (Montasser et al., 2020). Montasser et al. (2021) moreover use the algorithm from (Montasser et al., 2019) to get a sample complexity of $\tilde{O}\left(\frac{\mathsf{VC}(\mathcal{H})\mathsf{VC}^{*2}(\mathcal{H}) + \log(1/\delta)}{\epsilon}\right)$ and query complexity of $\tilde{O}(2^{\mathsf{VC}(\mathcal{H})^3 \mathsf{VC}^*(\mathcal{H})^2 \log^2(\mathsf{VC}^*(\mathcal{H}))}\mathsf{Lit}(\mathcal{H}))$. Finally, they extend their results to the agnostic setting and derive lower bounds. As in the setting with having only access to the example oracle, different notions of robustness have vastly different implications in terms of robust learnability of certain concept classes. Whenever relevant, we will draw a thorough comparison in the next sections between our work and that of Montasser et al. (2021).

## 2   Problem Set Up

We work in the PAC learning framework (see Appendix A.1), with the distinction that a robust risk function is used instead of the standard risk. We will study metric spaces $(\mathcal{X}_n, d)$ of input dimension $n$ with a perturbation budget function $\rho : \mathbb{N} \to \mathbb{R}$ defining the perturbation region $B_\rho(x) := \{z \in \mathcal{X}_n \mid d(x, z) \leq \rho(n)\}$. When the input space is the boolean hypercube $\mathcal{X}_n = \{0, 1\}^n$, the metric is the Hamming distance.

We use the exact-in-the-ball robust risk, which is defined w.r.t. a hypothesis $h$, target $c$ and distribution $D$ as the probability $\mathsf{R}_\rho^D(h, c) := \Pr_{x \sim D} (\exists z \in B_\rho(x) . c(z) \neq h(z))$ that $h$ and $c$ disagree in the perturbation region. On the other hand, the constant-in-the-ball robust risk is defined as $\Pr_{x \sim D} (\exists z \in B_\rho(x) . c(x) \neq h(z))$. Note that it is possible to adapt the latter to a joint distribution on the input and label spaces, but that there is an implicit *realizability assumption* in the former as the prediction on perturbed points' labels are compared to the ground truth $c$. We emphasize that choosing a robust risk function should depend on the learning problem at hand. The constant-in-the-ball notion of robustness requires a certain form of *stability*: the hypothesis should be correct on a random example and not change label in the perturbation region; this robust risk function may be more appropriate in settings with a strong margin assumption. In contrast, the exact-in-the-ball notion of robustness speaks to the *fidelity* of the hypothesis to the ground truth, and may be more suitable when a considerable portion of the probability mass is in the vicinity of the decision boundary. Diochnos et al. (2018); Dreossi et al. (2019); Gourdeau et al. (2021); Pydi and Jog (2021) offer a thorough comparison between different notions of robustness.

In the face of the impossibility or hardness of robustly learning certain concept classes, either through statistical or computational limitations, it is natural to study whether these issues can be circumvented by giving more power to the learner. The $\lambda$-local membership query ($\lambda$-LMQ) set up of Awasthi et al. (2013), which is formally defined in Appendix A.3, allows the learner to query the label of points that are at distance at most $\lambda$ from a sample $S$ drawn randomly from $D$. Inspired by this learning model, we define the $\lambda$-local equivalence query ($\lambda$-LEQ) model where, for a point $x$ in a sample $S$ drawn from the underlying distribution $D$, the learner is allowed to query an oracle that returns

---

[3]I.e., there exists a hypothesis that has zero constant-in-the-ball robust loss.

whether $h$ agrees with the ground truth $c$ in the ball $B_\lambda(x)$ of radius $\lambda$ around $x$.[4] If they disagree, a counterexample in $B_\lambda(x)$ is returned as well. Clearly, by setting $\lambda = n$, we recover the EQ oracle.[5] Note moreover that when $\lambda = \rho$, this is equivalent to querying the (exact-in-the-ball) robust loss around a point. We will show a separation result for robust learning algorithms between models that only allow random examples and ones that allow random examples and access to LEQ.

**Definition 1** ($\lambda$-LEQ Robust Learning). *Let $\mathcal{X}_n$ be the instance space, $\mathcal{C}$ a concept class over $\mathcal{X}_n$, and $\mathcal{D}$ a class of distributions over $\mathcal{X}_n$. We say that $\mathcal{C}$ is $\rho$-robustly learnable using $\lambda$-local equivalence queries with respect to distribution class, $\mathcal{D}$, if there exists a learning algorithm, $\mathcal{A}$, such that for every $\epsilon > 0$, $\delta > 0$, for every distribution $D \in \mathcal{D}$ and every target concept $c \in \mathcal{C}$, the following hold:*[6]

1. *$\mathcal{A}$ draws a sample $S$ of size $m = poly(n, 1/\delta, 1/\epsilon)$ using the example oracle $\mathsf{EX}(c, D)$*

2. *Each query made by $\mathcal{A}$ at $x \in S$ and for a candidate hypothesis $h$ to $\lambda$-LEQ either confirms that $c$ and $h$ coincide on $B_\lambda(x)$ or returns $z \in B_\lambda(x)$ such that $c(z) \neq h(z)$. $\mathcal{A}$ is allowed to update $h$ after seeing a counterexample*

3. *$\mathcal{A}$ outputs a hypothesis $h$ that satisfies $\mathsf{R}_\rho^D(h, c) \leq \epsilon$ with probability at least $1 - \delta$*

4. *The running time of $\mathcal{A}$ (hence also the number of oracle accesses) is polynomial in $n$, $1/\epsilon$, $1/\delta$ and the output hypothesis $h$ is polynomially evaluable.*

We remark that this model evokes the online learning setting, where the learner receives counterexamples after making a prediction, but with a few key differences. Contrary to the online setting (and the exact learning framework with MQ and EQ), there is an underlying distribution with which the performance of the hypothesis is evaluated in both the LMQ and LEQ models. Moreover, in online learning, when receiving a counterexample, the only requirement is that there is a concept that correctly classifies all the data given to the learner up until that point, and so the counterexamples can be given in an *adversarial* fashion, in order to maximize the regret. However, both the LMQ and LEQ models require that a target concept be chosen a priori. Note though that the LEQ oracle can give any counterexample for the robust loss at a given point.

In practice, one always has to find a way to approximately implement oracles studied in theory. A possible way to generate counterexamples with respect to the exact-in-the-ball notion of robustness is as follows. Suppose that there is an adversary that can generate points $z \in B_\rho(x)$ such that $h(z) \neq c(z)$. Provided such an adversary can be simulated, there is a way to (imperfectly) implement the LEQ oracle in practice.

Both the LMQ and LEQ models are particularly well-suited for the standard and exact-in-the-ball risks, as they address *information-theoretic* limitations of learning with random examples only. On the other hand, while information-theoretic limitations of robust learning with respect to the *constant-in-the-ball* notion of robustness arise when the perturbation function $\mathcal{U}$ is unknown to the learner, *computational* obstacles can also occur even when the definition of $\mathcal{U}$ is available. Indeed, determining whether the hypothesis changes label in the perturbation region could be intractable. In these cases, the Perfect Attack Oracle of Montasser et al. (2021) can be used to remedy these limitations for robust learning with respect to the constant-in-the-ball robust risk. Crucially, in their setting, counterexamples could have a different label to the ground truth: a counterexample $z \in \mathcal{U}(x)$ for $x$ is such that $h(z) \neq c(x)$, not necessarily $h(z) \neq c(z)$. This could compromise the standard accuracy of the hypothesis (see e.g., Tsipras et al. (2019) for a learning problem where robustness and accuracy are at odds). Finally, an LMQ analogue for the constant-in-the-ball risk is not needed: the only information we need for a perturbed point $z \in B_\rho(x)$ is the label of $x$ (given by the example oracle) and $h(z)$. Given that one of the requirements of PAC learning is that the hypothesis is efficiently evaluatable, we can easily compute $h(z)$.

---

[4]Similarly to $\rho$, we implicitly consider $\lambda$ as a function of the input dimension $n$. It is also possible to extend this definition to an arbitrary perturbation function $\mathcal{U} : \mathcal{X} \to 2^{\mathcal{X}}$.

[5]This is evidently not the case for the Perfect Attack Oracle of Montasser et al. (2021).

[6]We implicitly assume that a concept $c \in \mathcal{C}$ can be represented in size polynomial in $n$, where $n$ is the input dimension; otherwise a parameter $size(c)$ can be introduced in the sample and query complexity requirements.

# 3 Distribution-Free Robust Learning with Local Equivalence Queries

In this section, we show that having access to a local equivalence query oracle can guarantee the efficient *distribution-free* robust learnability of certain concept classes. We start with a negative result which shows that for a wide variety of concept classes, if $\lambda < \rho$, then *distribution-free* robust learnability is impossible with $\mathsf{EX} + \lambda\text{-}\mathsf{LEQ}$ – regardless of how many queries are allowed. However, the regime $\lambda = \rho$, which implies giving similar power to the learner as the adversary, enables robust learnability guarantees. Indeed, Section 3.2 exhibits upper bounds on sample sizes that will guarantee *robust* generalization. These bounds are logarithmic in the size of the hypothesis class (finite case) and linear in the *robust* VC dimension of a concept class (infinite case). Section 3.3 draws a comparison between our framework and the online learning setting, and exhibits robustly consistent learners. Section 3.4 studies conjunctions and presents a robust learning algorithm that is *both* statistically and computationally efficient. Finally, Section 3.5 looks at linear classifiers in the discrete and continuous cases, and adapts the Winnow and Perceptron algorithms to both settings.

## 3.1 Impossibility of Distribution-Free Robust Learning When $\lambda < \rho$

We start with a negative result, saying that whenever the local query radius is strictly smaller than the adversary's budget, monotone conjunctions are not distribution-free robustly learnable, which is in contrast to the standard PAC setting where guarantees hold *for any distribution*. Note that our result goes beyond efficiency: no query can distinguish between two potential targets. Choosing the target uniformly at random lower bounds the expected robust risk, and hence renders robust learning impossible in this setting. The proof of this theorem can be found in Appendix C.1.

**Theorem 2.** *For locality and robustness parameters $\lambda, \rho \in \mathbb{N}$ with $\lambda < \rho$, monotone conjunctions (and any superclass) are not distribution-free $\rho$-robustly learnable with access to a $\lambda$-$\mathsf{LEQ}$ oracle.*

The result holds for monotone conjunctions and all superclasses (e.g., decision lists and halfspaces), but, in fact, we can generalize this reasoning for any concept class that has a certain form of stability: if we can find concepts $c_1$ and $c_2$ in $\mathcal{C}$ and points $x, x' \in \mathcal{X}$ such that $c_1$ and $c_2$ agree on $B_\lambda(x)$ but disagree on $x'$, then if $\lambda < \rho$, the concept class $\mathcal{C}$ is not distribution-free $\rho$-robustly learnable with access to a $\lambda$-$\mathsf{LEQ}$ oracle. It suffices to "move" the center of the ball $x$ until we find a point in the set $B_\rho(x) \setminus B_\lambda(x)$ where $c_1$ and $c_2$ disagree, which is guaranteed to happen by the existence of $x'$.

## 3.2 General Sample Complexity Bounds for Robustly-Consistent Learners

In this section, we show that we can derive sample complexity upper bounds for *robustly* consistent learners, i.e., learning algorithms that return a *robust* loss of zero on a training sample. Note that, crucially, the exact-in-the-ball notion of robustness and its realizability imply that any robust ERM algorithm will achieve zero empirical robust loss on a given training sample. As we will see in the next sections, the challenge is to find a *robustly* consistent learning algorithm that uses queries to $\rho$-$\mathsf{LEQ}$. The first bound is for finite classes, where the dependency is logarithmic in the size of the hypothesis class. The proof is a simple application of Occam's razor and is included in Appendix C.2 for completeness. The reasoning is similar to Bubeck et al. (2019).

**Lemma 3.** *Let $\mathcal{C}$ be a concept class and $\mathcal{H}$ a hypothesis class. Any $\rho$-robust ERM algorithm using $\mathcal{H}$ on a sample of size $m \geq \frac{1}{\epsilon}\left(\log|\mathcal{H}_n| + \log\frac{1}{\delta}\right)$ is a $\rho$-robust learner for $\mathcal{C}$.*

For the infinite case, we cannot immediately use the VC dimension as a tool for bounding the sample complexity of robust learning. To this end, we define the *robust* VC dimension of a concept class, which is the VC dimension of the class of functions representing the $\rho$-expansion of the error region between any possible target and hypothesis. This definition is analogous to the adversarial VC dimension defined by Cullina et al. (2018) for the constant-in-the-ball definition of robustness.

**Definition 4** (Robust VC dimension). *Given a target concept class $\mathcal{C}$, a hypothesis class $\mathcal{H}$ and a robustness parameter $\rho$, the robust VC dimension is defined as $\mathsf{RVC}_\rho(\mathcal{C}, \mathcal{H}) = \mathsf{VC}((\mathcal{C} \oplus \mathcal{H})_\rho)$, where $(\mathcal{C} \oplus \mathcal{H})_\rho = \{(c \oplus h)_\rho : x \mapsto \mathbf{1}[\exists z \in B_\rho(x) . c(z) \neq h(z)] \mid c \in \mathcal{C}, h \in \mathcal{H}\}$. Whenever $\mathcal{C} = \mathcal{H}$, we simply write $\mathsf{RVC}_\rho(\mathcal{C})$.*

We now show that we can use the robust VC dimension to upper bound the sample complexity of robustly-consistent learning algorithms. We will use this result in Section 3.5 when dealing with an infinite concept class: halfspaces on $\mathbb{R}^n$.

**Lemma 5.** *Let $\mathcal{C}$ be a concept class and $\mathcal{H}$ a hypothesis class. Any $\rho$-robust ERM algorithm using $\mathcal{H}$ on a sample of size $m = \Omega\left(\frac{1}{\epsilon}\left(\mathsf{RVC}_\rho(\mathcal{C},\mathcal{H})\log(1/\epsilon) + \log\frac{1}{\delta}\right)\right)$ is a $\rho$-robust learner for $\mathcal{C}$.*

*Proof Sketch of Lemma 5.* The proof is very similar to the VC dimension upper bound in PAC learning. The main distinction is that, instead of looking at the error region of the target and any function in $\mathcal{H}$, we look at its $\rho$-expansion. Namely, let the target $c \in \mathcal{C}$ be fixed and, for $h \in \mathcal{H}$, consider the function $(c \oplus h)_\rho : x \mapsto \mathbf{1}[\exists z \in B_\rho(x) \ . \ c(z) \neq h(z)]$ and define a new concept class $\Delta_{c,\rho}(\mathcal{H}) = \{(c \oplus h)_\rho \mid h \in \mathcal{H}\}$. It is easy to show that $\mathsf{VC}(\Delta_{c,\rho}(\mathcal{H})) \leq \mathsf{RVC}_\rho(\mathcal{C},\mathcal{H})$, as any sign pattern achieved on the LHS can be achieved on the RHS. The rest of the proof follows from the definition of an $\epsilon$-net and the bound on the growth function of $\Delta_{c,\rho}(\mathcal{H})$; see Appendix C.3 for details. $\qquad\square$

*Remark 6.* Note that, as $\rho(n)/n$ tends to $1$, we move towards the exact and online learning settings, and the underlying distribution becomes less important. In this case, the robust VC dimension starts to decrease. Indeed, say if $\rho = n$, then $(\mathcal{C} \oplus \mathcal{C})_\rho$ only contains the constant functions $0$ and $1$. We thus only need a single example to query the LEQ oracle (which has become the EQ oracle). However, this comes at a cost: the *query complexity* upper bounds presented in the next sections could be tight. Understanding the behaviour of the robust VC dimension as a function of $\rho$ and deriving joint sample and query complexity bounds are both avenues for future research.

## 3.3 Query Complexity Bounds Using Online Learning Results

In the previous section, we derived sample complexity upper bounds for robustly consistent learners. The challenge is thus to create algorithms that perform robust empirical risk minimization, as we are operating in the realizable setting. We begin by showing that, if one can ignore computational limitations, then online learning results can be used to guarantee robust learnability. We recall the online learning setting in Appendix A.5. We denote by $\mathsf{Lit}(\mathcal{C})$ the Littlestone dimension of a concept class $\mathcal{C}$, which is defined in Appendix A.4 and appears in the query complexity bound in the theorem below, whose proof can be found in Appendix C.4.

**Theorem 7.** *A concept class $\mathcal{C}$ is $\rho$-robustly learnable with the Standard Optimal Algorithm (Littlestone, 1988) using the EX and $\rho$-LEQ oracles with sample and query complexity $m(n,\epsilon,\delta) = \Omega\left(\frac{1}{\epsilon}\left(\mathsf{RVC}_\rho(\mathcal{C})\log(1/\epsilon) + \log\frac{1}{\delta}\right)\right)$ and $r(n,\epsilon,\delta) = m(n,\epsilon,\delta)\cdot\mathsf{Lit}(\mathcal{C})$, respectively. Furthermore, if $\mathcal{C}$ is a finite concept class on $\{0,1\}^n$, then $\mathcal{C}$ is $\rho$-robustly learnable with sample and query complexity $m(n,\epsilon,\delta) = \frac{1}{\epsilon}\left(\log(|\mathcal{C}|) + \log\frac{1}{\delta}\right)$ and $r(n,\epsilon,\delta,\rho) = m(n,\epsilon,\delta)\cdot\mathsf{Lit}(\mathcal{C})$.*

Of course, some concept classes, e.g., thresholds, have infinite Littlestone dimension, so our bounds are not useful in these settings. In Section 3.5, we study distributional assumptions that give reasonable query upper bounds for linear classifiers, using the theorem below. It exhibits a query upper bound for robustly learning with an online algorithm $\mathcal{A}$ with a given mistake upper bound $M$. This is moreover particularly useful in case $M$ is polynomial in the input dimension and $\mathcal{A}$ is *computationally* efficient (which is not the case for the Standard Optimal Algorithm in Theorem 7).

**Lemma 8.** *Let $\mathcal{C}$ be a concept class learnable in the online setting with mistake bound $M(n)$. Then $\mathcal{C}$ is $\rho$-robustly learnable using the EX and $\rho$-LEQ oracles with sample complexity $m(n,\epsilon,\delta) = \frac{1}{\epsilon}\left(\mathsf{RVC}_\rho(\mathcal{H},C) + \log\frac{1}{\delta}\right)$ and query complexity $r(n,\epsilon,\delta) = m(n,\epsilon,\delta)\cdot M(n)$.*

*Proof.* The sample complexity bound is obtained from Lemma 5 and, for each point in the sample, a query to LEQ can either return a robust loss of $0$ or $1$ and give a counterexample. Since the mistake bound is $M(n)$, we have a query upper bound of $r = m \cdot M$, as required. $\qquad\square$

## 3.4 Improved Query Complexity Bounds: Conjunctions

In this section, we show how to improve the query upper bound from the previous section in the special case of conjunctions. Moreover, the algorithm used to robustly learn conjunctions is both statistically and *computationally* efficient, which is not the case of the Standard Optimal Algorithm. The proof of the following theorem can be found in Appendix C.5.

**Theorem 9.** *The class CONJUNCTIONS is efficiently $\rho$-robustly learnable in the distribution-free setting using the EX and $\rho$-LEQ oracles with at most $O\left(\frac{1}{\epsilon}\left(n + \log\frac{1}{\delta}\right)\right)$ random examples and $O\left(\frac{1}{\epsilon}\left(n + \log\frac{1}{\delta}\right)\right)$ queries to $\rho$-LEQ.*

Note that the query upper bound that we get is of the form $m + M$, as opposed to $m \cdot M$ from Lemma 5 (where $m$ is the sample complexity and $M$ the mistake bound). This is because we have adapted the PAC learning algorithm for conjunctions to our setting. Any update to its hypothesis will not affect the consistency of previously queried points with robust loss of zero, and thus once zero robust loss is achieved on a point, it does not need to be queried again.

## 3.5 Linear Classifiers

In this section, we derive sample and query complexity upper bounds for restricted subclasses of linear classifiers. We start with linear classifiers on $\{0,1\}^n$ with bounded weights, and continue with linear classifiers on $\mathbb{R}^n$ with a margin condition. We use the well-known Winnow and Perceptron algorithms. Note that the robustness threshold[7] of linear classifiers on $\{0,1\}^n$ *without* access to the LEQ oracle remains an open problem (Gourdeau et al., 2022).

Let $\mathsf{LTF}_{\{0,1\}^n}^W$ be the class of linear threshold functions on $\{0,1\}^n$ with integer weights such that the sum of the absolute values of the weights and the bias is bounded above by $W$. We have the following theorem, whose proof relies on bounding the size of $\mathsf{LTF}_{\{0,1\}^n}^W$ and using the mistake bound for Winnow (Littlestone, 1988). The proof can be found in Appendix C.6.

**Theorem 10.** *The class* $\mathsf{LTF}_{\{0,1\}^n}^W$ *is $\rho$-robustly learnable with access to the* EX *and $\rho$-LEQ oracles by using the Winnow algorithm with sample complexity* $m(n, \epsilon, \delta) = O\left(\frac{1}{\epsilon}\left(n + \min\{n, W\}\log(W + n) + \log\frac{1}{\delta}\right)\right)$ *and query complexity* $O(m(n, \epsilon, \delta) \cdot W^2 \log(n))$.

Now, we derive sample and query complexity upper bounds for the robust learnability of linear classifiers $\mathsf{LTF}_{\mathbb{R}^n}$ on $\mathbb{R}^n$. Note that, unlike in previous results, the distribution family is restricted to guarantee the existence of a margin for each concept and distribution pair, and so we cannot guarantee distribution-free robust learning in this case. This is because the Littlestone dimension of thresholds, and thus halfspaces, is infinite if there are no distributional assumptions on this concept class. We remark that whenever the margin $\gamma$ is greater than $\rho/2$, the constant and exact-in-the-ball notions of robustness could coincide,[8] in which case the results from (Diakonikolas et al., 2020; Montasser et al., 2021) apply. However, unlike in Diakonikolas et al. (2020); Montasser et al. (2021), under our notion of robustness, if $\gamma < \rho/2$, we may still be in the realizable setting (there exists at least one concept that is robustly consistent with the data), while when considering the constant-in-the-ball risk, we are necessarily in the non-realizable/agnostic setting. As mentioned earlier, guarantees obtained in the latter do not necessarily translate to the former. The full proof of the theorem below appears in Appendix C.7.

**Theorem 11.** *Fix constants $B, \gamma > 0$. Let $\mathcal{L} = \{(c, D) \mid c \in \mathsf{LTF}_{\mathbb{R}^n}, D \in \mathcal{D}\}$ be a family of halfspace and distribution pairs, where each pair $(c, D)$ with $c(x) = a^\top x + a_0$ is such that if $x \in supp(D)$, then (i) $\|x\|_2 \le B$ and (ii) $\gamma \le \frac{c(x)(a^\top x)}{\|x\|_2}$, i.e., $D$ has support bounded by $B$ and induces a margin of $\gamma$ w.r.t. $c$. Let the adversary's budget be measured by the $\ell_2$ norm. Then, $\mathcal{L}$ is $\rho$-robustly learnable using the EX and $\rho$-LEQ oracles with sample complexity $m = O(\frac{1}{\epsilon}(n^3 + \log(1/\delta)))$ and query complexity $r = \frac{mB^2}{\gamma^2}$. Note that this is query-efficient if $\frac{B^2}{\gamma^2} = poly(n)$.*

*Proof Sketch.* The first step is to derive the sample complexity bound. To this end, we use Lemma 5 and bound the robust VC dimension of linear classifiers on $\mathbb{R}^n$. We do this using a result of Goldberg and Jerrum (1995) (Theorem 28 in Appendix C.7), which bounds the VC dimension of concept classes expressible as boolean combinations of polynomial inequalities. We first express the $\rho$-expansion of the error region, i.e., the robust loss, between two linear classifiers as a first order logical formula $\psi$ over the reals where the atomic predicates are polynomial inequalities. We then use the quantifier-elimination method from Renegar (1992) to transform $\psi$ into a quantifier-free formula $\varphi$. This method allows us to show an upper bound on the number of atomic predicates, their degree, and the number of variables in $\varphi$. We can apply the result of Goldberg and Jerrum (1995) on $\varphi$ to get a robust VC dimension of $O(n^3)$.

The second step is to derive the query upper bound, which follows from Lemma 8 and the mistake bound for the Perceptron algorithm, which appears in Appendix B. $\square$

---

[7]With respect to the exact-in-the-ball definition of robustness.
[8]Given that the choice of target implies constant-in-the-ball realizability.

# 4 A Local Membership Query Lower Bound for Conjunctions

In this section, we show that the amount of data needed to $\rho$-robustly learn conjunctions under the uniform distribution has an exponential dependence on the adversary's budget $\rho$ when the learner only has access to the EX and LMQ oracles. Here, the lower bound on the sample drawn from the example oracle is $2^\rho$, which is the same as the lower bound for *monotone* conjunctions derived in Gourdeau et al. (2022), and the local membership query lower bound is $2^{\rho-1}$. The result relies on showing there there exists a family of conjunctions that remain indistinguishable from each other on any sample of size $2^\rho$ and any sequence of $2^{\rho-1}$ LMQs with constant probability.

**Theorem 12.** *Fix a monotone increasing robustness function $\rho : \mathbb{N} \to \mathbb{N}$ satisfying $2 \leq \rho(n) \leq n/4$ for all $n$. Then, for any query radius $\lambda$, any $\rho(n)$-robust learning algorithm for the class* CONJUNCTIONS *with access to the* EX *and* $\lambda$-LMQ *oracles has joint sample and query complexity lower bounds of $2^\rho$ and $2^{\rho-1}$ under the uniform distribution.*

*Proof.* Let $D$ be the uniform distribution and WLOG let $\rho \geq 2$. Fix two disjoint sets $I_1$ and $I_2$ of $2\rho$ indices in $[n]$, which will be the set of variables appearing in potential target conjunctions $c_1$ and $c_2$, respectively (i.e., their support). We have $2^{4\rho}$ possible pairs of such conjunctions, as each variable can appear as a positive or negative literal.

Let us consider a randomly drawn sample $S$ of size $2^\rho$. We will first consider what happens when all the examples in $S$ and the queried inputs $S'$ are negatively labelled. Each negative example $x \in S$ allows us to remove at most $2^{2\rho+1}$ pairs from the possible set of pairs of conjunctions, as each component $x_{I_1}$ and $x_{I_2}$ removes at most one conjunction from the possible targets. By the same reasoning, each LMQ that returns a negative example can remove at most $2^{2\rho+1}$ pairs of conjunctions. Note that the parameter $\lambda$ is irrelevant in this setting as each LMQ can only test one concept pair. Thus, after seeing any random sample of size $2^\rho$ and querying any $2^{\rho-1}$ points, there remains

$$\frac{2^{4\rho} - 2^{3\rho+1} - 2^{3\rho}}{2^{4\rho}} \geq 1/4 \tag{1}$$

of the initial conjunction pairs that label all points in $S$ and $S'$ negatively. Then, fixing $S, S'$ and choosing a pair $(c_1, c_2)$ of possible target conjunctions uniformly at random and then choosing $c$ uniformly at random between the two gives at least a $1/4$ chance that $S$ and $S'$ only contain negative examples (both conjunctions are consistent with this).

Moreover, note that any two conjunctions in a pair will have a robust risk lower bounded by $15/32$ against each other under the uniform distribution (see Lemma 23 in Appendix B). Thus, any learning algorithm $\mathcal{A}$ with LMQ query budget $m' = 2^{\rho-1}$ and strategy $\sigma : (\{0,1\}^n \times \{0,1\})^m \to (\{0,1\}^n \times \{0,1\})^{m'}$ (note that the queries can be adaptive) can do no better than to guess which of $c_1$ or $c_2$ is the target if they are both consistent on the augmented sample $S \cup \sigma(S)$, giving an expected robust risk lower bounded by a constant. Letting $\mathcal{E}$ be the event that all points in both $S$ and $\sigma(S)$ are labelled zero, we get

$$\mathbb{E}_{c,S}\left[\mathsf{R}_\rho^D(\mathcal{A}(S \cup \sigma(S)), c)\right] \geq \Pr_{c,S}(\mathcal{E}) \, \mathbb{E}_{c,S}\left[\mathsf{R}_\rho^D(\mathcal{A}(S \cup \sigma(S)), c) \mid \mathcal{E}\right] \quad \text{(Law of Total Expectation)}$$

$$\geq \frac{1}{4} \, \mathbb{E}_{c,S}\left[\mathsf{R}_\rho^D(\mathcal{A}(S \cup \sigma(S)), c) \mid \mathcal{E}\right] \quad \text{(Equation 1)}$$

$$= \frac{1}{4} \cdot \frac{1}{2} \, \mathbb{E}_S\left[\mathsf{R}_\rho^D(\mathcal{A}(S \cup \sigma(S)), c_1) + \mathsf{R}_\rho^D(\mathcal{A}(S \cup \sigma(S)), c_2) \mid \mathcal{E}\right]$$
$$\text{(Random choice of } c)$$

$$\geq \frac{1}{8} \, \mathbb{E}_S\left[\mathsf{R}_\rho^D(c_1, c_2) \mid \mathcal{E}\right] \quad \text{(Lemma 22)}$$

$$> \frac{1}{8} \cdot \frac{15}{32} \quad \text{(Lemma 23)}$$

$$= \frac{15}{256} \, ,$$

which completes the proof. $\qquad\qquad\square$

We use the term *robustness threshold* from Gourdeau et al. (2021) to denote an adversarial budget function $\rho : \mathbb{N} \to \mathbb{R}$ of the input dimension $n$ such that, if the adversary is allowed perturbations

of magnitude $\rho(n)$, then there exists a sample-efficient $\rho(n)$-robust learning algorithm, and if the adversary's budget is $\omega(\rho(n))$, then there does not exist such an algorithm. Robustness thresholds are distribution-dependent when the learner only has access to the example oracle EX, as seen in (Gourdeau et al., 2021, 2022). Now, since the local membership query lower bound above has an exponential dependence on $\rho$, any perturbation budget $\omega(\log n)$ will require a sample and query complexity that is superpolynomial in $n$, giving the following corollary.

**Corollary 13.** *The robustness threshold of the class* CONJUNCTIONS *under the uniform distribution with access to* EX *and an* LMQ *oracle is* $\Theta(\log(n))$.

The robustness threshold above is the same as when only using the EX oracle (Gourdeau et al., 2021). Finally, since decision lists and halfspaces both subsume conjunctions, the lower bound of Theorem 12 also holds for these classes.

## 5   Conclusion

We have shown that local membership queries do not change the robustness threshold of conjunctions, or any superclass, under the uniform distribution. However, access to a $\rho$-local *equivalence* query oracle allows us to develop robust ERM algorithms. We have introduced the notion of robust VC dimension to determine sample complexity bounds and have used online learning results to derive query complexity bounds. We have moreover adapted the PAC learning algorithm for conjunctions for this setting and have greatly improved its query complexity compared to the general case. Finally, we have studied halfspaces, both in the boolean hypercube and continuous settings. The latter is, to our knowledge, the first robust learning algorithm with respect to the exact-in-the-ball notion of robustness for a non-trivial concept class in $\mathbb{R}^n$. Overall, we have shown that the LEQ oracle is *essential* to ensure the *distribution-free* robust learning of commonly studied concept classes in our setting. Note that this is in contrast with standard PAC learning with the EX and EQ oracles, where equivalence queries don't give more power to learner.

We finally outline various avenues for future research:

1. Can we give a more fine-grained picture of the sample and query complexity tradeoff outlined in Remark 6, e.g., by improving LEQ query upper bounds when $\rho$ is small?

2. Can we derive sample and query lower bounds for robust learning with an LEQ oracle?

3. The LMQ lower bound from Section 4 was derived for conjunctions. The technique does not work for monotone conjunctions.[9] Can we get a similar LMQ lower bound where the dependence on $\rho$ is exponential for monotone conjunctions, or it is possible to robustly learn them with $o(2^\rho)$ local membership queries?

## Acknowledgements

MK and PG received funding from the ERC under the European Union's Horizon 2020 research and innovation programme (FUN2MODEL, grant agreement No. 834115).

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
