# A Preliminaries

## A.1 The PAC Framework

**Definition 14** (PAC Learning, Valiant (1984))**.** *Let $\mathcal{C}_n$ be a concept class over $\mathcal{X}_n$ and let $\mathcal{C} = \bigcup_{n \in \mathbb{N}} \mathcal{C}_n$. We say that $\mathcal{C}$ is PAC learnable using hypothesis class $\mathcal{H}$ and sample complexity function $p(\cdot, \cdot, \cdot, \cdot)$ if there exists an algorithm $\mathcal{A}$ that satisfies the following: for all $n \in \mathbb{N}$, for every $c \in \mathcal{C}_n$, for every $D$ over $\mathcal{X}_n$, for every $0 < \epsilon < 1/2$ and $0 < \delta < 1/2$, if whenever $\mathcal{A}$ is given access to $m \geq p(n, 1/\epsilon, 1/\delta, size(c))$ examples drawn i.i.d. from $D$ and labeled with $c$, $\mathcal{A}$ outputs a polynomially evaluatable $h \in \mathcal{H}$ such that with probability at least $1 - \delta$,*

$$\Pr_{x \sim D} (c(x) \neq h(x)) \leq \epsilon .$$

*We say that $\mathcal{C}$ is statistically efficiently PAC learnable if $p$ is polynomial in $n, 1/\epsilon, 1/\delta$ and $size(c)$, and computationally efficiently PAC learnable if $\mathcal{A}$ runs in polynomial time in $n, 1/\epsilon, 1/\delta$ and $size(c)$.*

The setting where $\mathcal{C} = \mathcal{H}$ is called *proper learning*, and *improper learning* otherwise. The PAC setting where the guarantees hold for any distribution is called *distribution-free*.

## A.2 Robust Learnability

**Definition 15** (Efficient Robust Learnability, Gourdeau et al. (2021))**.** *Fix a function $\rho : \mathbb{N} \to \mathbb{N}$. We say that an algorithm $\mathcal{A}$ efficiently $\rho$-robustly learns a concept class $\mathcal{C}$ with respect to distribution class $\mathcal{D}$ if there exists a polynomial $poly(\cdot, \cdot, \cdot, \cdot)$ such that for all $n \in \mathbb{N}$, all target concepts $c \in \mathcal{C}_n$, all distributions $D \in \mathcal{D}_n$, and all accuracy and confidence parameters $\epsilon, \delta > 0$, if $m \geq poly(n, 1/\epsilon, 1/\delta, size(c))$, whenever $\mathcal{A}$ is given access to a sample $S \sim D^m$ labelled according to $c$, it outputs a polynomially evaluable function $h : \{0,1\}^n \to \{0,1\}$ such that $\Pr_{S \sim D^m} (R_\rho(h, c) < \epsilon) > 1 - \delta$.*

## A.3 Local Membership Queries and Robust Learning

We recall the formal definition of the LMQ model from (Awasthi et al., 2013), but where we have changed the standard risk to the robust risk. Here, given a sample $S$ drawn from the example oracle, a membership query for a point $x$ is $\lambda$-local if there exists $x' \in S$ such that $x \in B_\lambda(x')$.

**Definition 16** ($\lambda$-LMQ Robust Learning)**.** *Let $\mathcal{X}$ be the instance space, $\mathcal{C}$ a concept class over $\mathcal{X}$, and $\mathcal{D}$ a class of distributions over $\mathcal{X}$. We say that $\mathcal{C}$ is $\rho$-robustly learnable using $\lambda$-local membership queries with respect to $\mathcal{D}$ if there exists a learning algorithm $\mathcal{A}$ such that for every $\epsilon > 0$, $\delta > 0$, for every distribution $D \in \mathcal{D}$ and every target concept $c \in \mathcal{C}$, the following hold:*

1. *$\mathcal{A}$ draws a sample $S$ of size $m = poly(n, 1/\delta, 1/\epsilon, size(c))$ using the example oracle $\mathsf{EX}(c, D)$*

2. *Each query $x'$ made by $\mathcal{A}$ to the $\mathsf{LMQ}$ oracle is $\lambda$-local with respect to some example $x \in S$*

3. *$\mathcal{A}$ outputs a hypothesis $h$ that satisfies $R_\rho^D(h, c) \leq \epsilon$ with probability at least $1 - \delta$*

4. *The running time of $\mathcal{A}$ (hence also the number of oracle accesses) is polynomial in $n, 1/\epsilon, 1/\delta, size(c)$ and the output hypothesis $h$ is polynomially evaluable.*

## A.4 Complexity Measures

For a more in-depth introduction to these concepts, we refer the reader to Mohri et al. (2012).

**Definition 17** (Shattering)**.** *Given a class of functions $\mathcal{F}$ from input space $\mathcal{X}$ to $\{0,1\}$, we say that a set $S \subseteq \mathcal{X}$ is shattered by $\mathcal{F}$ if all the possible dichotomies of $S$ (i.e., all the possible ways of labelling the points in $S$) can be realized by some $f \in \mathcal{F}$.*

**Definition 18** (VC Dimension)**.** *The $\mathsf{VC}$ dimension of a hypothesis class $\mathcal{H}$, denoted $\mathsf{VC}(\mathcal{H})$, is the size $d$ of the largest set that can be shattered by $\mathcal{H}$. If no such $d$ exists then $\mathsf{VC}(\mathcal{H}) = \infty$.*

**Definition 19** (Littlestone Tree)**.** *A Littlestone tree for a hypothesis class $\mathcal{H}$ on $\mathcal{X}$ is a complete binary tree $T$ of depth $d$ whose internal nodes are instances $x \in \mathcal{X}$. Each edge is labeled with $-$ or*

+ *and corresponds to the potential labels of the parent node. Each path from the root to a leaf must be consistent with some* $h \in \mathcal{H}$*, i.e. if* $x_1, \dots, x_d$ *with labelings* $y_1, \dots, y_d$ *is a path in* $T$*, there must exist* $h \in \mathcal{H}$ *such that* $h(x_i) = y_i$ *for all* $i$*.*

**Definition 20** (Littlestone Dimension). *The Littlestone dimension of a hypothesis class* $\mathcal{H}$*, denoted* $\mathsf{Lit}(\mathcal{H})$*, is the depth* $d$ *of the largest Littlestone tree for* $\mathcal{H}$*. If no such* $d$ *exists then* $\mathsf{Lit}(\mathcal{H}) = \infty$*.*

### A.5 Online Learning

In online learning, the learner is given access to examples *sequentially*. At each time step, the learner receives an example $x$, predicts its label using its hypothesis $h$, receives the true label $y$ and updates its hypothesis if $h(x) \neq y$. A fundamental difference between PAC learning and online learning is that, in the latter, there are no distributional assumptions. Examples can be given adversarially, and the performance of the learner is evaluated with respect to the number of mistakes it makes compared to the ground truth.

**Definition 21** (Mistake Bound). *For a given hypothesis class* $\mathcal{C}$ *and instance space* $\mathcal{X} = \bigcup_n \mathcal{X}_n$*, we say that an algorithm* $\mathcal{A}$ *learns* $\mathcal{C}$ *with mistake bound* $M$ *if* $A$ *makes at most* $M$ *mistakes on any sequence of samples consistent with a concept* $c \in \mathcal{C}$*. In the mistake bound model, we usually require that* $M$ *be polynomial in* $n$ *and size$(c)$.*

We now recall the Standard Optimal Algorithm (Littlestone, 1988), which has a mistake bound $M = \mathsf{Lit}(\mathcal{C})$ when given concept class $\mathcal{C}$.

---

**Algorithm 1** Standard Optimal Algorithm from Littlestone (1988)

---

**Input:** A hypothesis class $\mathcal{H}$
    **for** $t = 1, 2, \dots$ **do**
        Receive example $x_t$
        $V_t^{(b)} \leftarrow \{h \in V_t \mid h(x_t) = b\}$
        $\hat{y}_t = \arg\max_b \mathsf{Lit}(V_t^{(b)})$      ▷ Predict label acc. to subclass with larger Littlestone dimension
        Receive true label $y_t$
        $V_{t+1} \leftarrow V_t^{(y_t)}$
    **end for**

---

## B Useful Results

### B.1 Robust Risk Bounds

**Lemma 22** (Lemma 6 in Gourdeau et al. (2019)). *Let* $c_1, c_2 \in \{0,1\}^{\mathcal{X}}$ *and fix a distribution* $D$ *on* $\mathcal{X}$*. Then, for all* $h : \{0,1\}^n \to \{0,1\}$*,*

$$R_\rho^D(c_1, c_2) \leq R_\rho^D(c_1, h) + R_\rho^D(c_2, h) \ .$$

**Lemma 23** (Lemma 14 in Gourdeau et al. (2022)). *Under the uniform distribution, for any* $n \in \mathbb{N}$*, disjoint* $c_1, c_2 \in$ *MON-CONJ of even length* $3 \leq l \leq n/2$ *on* $\{0,1\}^n$ *and robustness parameter* $\rho = l/2$*, we have that* $\mathsf{R}_\rho^D(c_1, c_2)$ *is bounded below by a constant that can be made arbitrarily close to* $\frac{1}{2}$ *as* $l$ *(and thus* $\rho$*) increases.*

*Remark* 24. Note that the statement and proof of the above lemma remains unchanged if considering disjoint conjunctions, as opposed to monotone conjunctions.

### B.2 Mistake Bounds for Winnow and Perceptron

Now, we recall the mistake upper bound for Winnow in the special case of $\mathsf{LTF}_{\{0,1\}^n}^{W+}$, where the weights are positive integers[10] and the mistake bound for the Perceptron algorithm.

**Theorem 25** (Winnow). *The Winnow algorithm for learning the class* **LTF**$_{\{0,1\}^n}^{W+}$ *makes at most* $O(W^2 \log(n))$ *mistakes.*

---

[10]See https://www.cs.utexas.edu/ klivans/05f7.pdf for a full derivation.

**Theorem 26** (Mistake Bound for Perceptron, Margin Condition; Theorem 7.8 in Mohri et al. (2012)). *Let $\mathbf{x}_1, \ldots, \mathbf{x}_T \in \mathbb{R}^n$ be a sequence of $T$ points with $\|\mathbf{x}_t\| \leq r$ for all $1 \leq t \leq T$ for some $r > 0$. Assume that there exists $\gamma > 0$ and $\mathbf{v} \in \mathbb{R}^n$ such that for all $1 \leq t \leq T$, $\gamma \leq \frac{y_t(\mathbf{v} \cdot \mathbf{x}_t)}{\|\mathbf{v}\|}$. Then, the number of updates made by the Perceptron algorithm when processing $\mathbf{x}_1, \ldots, \mathbf{x}_T$ is bounded by $r^2/\gamma^2$.*

### B.3 Quantifier Elimination

**Theorem 27** (Theorem 1.2 in Renegar (1992)). *Let $\Psi$ be a formula in the first-order theory of the reals of the form*

$$(Q_1 x^{[1]} \in \mathbb{R}^{n_1}) \ldots (Q_\omega x^{[\omega]} \in \mathbb{R}^{n_\omega}) P(x^{[1]}, \ldots, x^{[n_\omega]}, y) ,$$

*with free variables $y = (y_1, \ldots, y_l)$, quantifiers $Q_i$ ($\exists$ or $\forall$) and quantifier-free Boolean formula $P(x^{[1]}, \ldots, x^{[n_\omega]}, y)$ with $m$ atomic predicates consisting of polynomial inequalities of degree at most $d$. There exists a quantifier elimination method which constructs a quantifier-free formula $\Phi$ of the form*

$$\bigvee_{i=1}^{I} \bigwedge_{j=1}^{J_i} (h_{ij}(y) \Delta_{ij} 0) ,$$

*where*

$$I \leq (md)^{2^{O(\omega)} l \prod_k n_k}$$
$$J_i \leq (md)^{2^{O(\omega)} \prod_k n_k}$$
$$\deg(h_{ij}) \leq (md)^{2^{O(\omega)} \prod_k n_k}$$
$$\Delta_{ij} \in \{\leq, \geq, =, \neq, >, <\} .$$

## C   Proofs from Section 3

### C.1   Proof of Theorem 2

*Proof.* Fix $\lambda, \rho \in \mathbb{N}$ such that $\lambda < \rho$, and consider the following monotone conjunctions: $c_1(x) = \bigwedge_{1 \leq i \leq \rho} x_i$ and $c_2(x) = \bigwedge_{1 \leq i \leq \rho+1} x_i$. Let $D$ be the distribution on $\{0, 1\}^n$ which puts all the mass on $\mathbf{0}$. Then, the target concept is drawn at random between $c_1$ and $c_2$. Now, $c_1$ and $c_2$ will both give all points in $B_\lambda(\mathbf{0})$ the label 0, so the learner has to choose a hypothesis that is consistent with both $c_1$ and $c_2$ (otherwise the robust risk is 1 and we are done). However, the learner has no way of distinguishing which of $c_1$ or $c_2$ is the target concept, while these two functions have a $\rho$-robust risk of 1 against each other under $D$. Formally,

$$
\begin{aligned}
\mathsf{R}_\rho^D(c_1, c_2) &= \Pr_{x \sim D} (\exists z \in B_\rho(x) \, . \, c_1(z) \neq c_2(z)) \\
&= \mathbf{1}[\exists z \in B_\rho(\mathbf{0}) \, . \, c_1(z) \neq c_2(z)] \\
&= 1 ,
\end{aligned}
\tag{2}
$$

where such $z = \mathbf{1}_\rho \mathbf{0}_{n-\rho}$. To lower bound the expected robust risk, letting $\mathcal{A}$ be any learning algorithm and $\mathcal{E}$ be the event that all points in a randomly drawn sample $S$ are all labeled 0, we have

$$
\begin{aligned}
\mathbb{E}_{c,S} \left[ \mathsf{R}_\rho^D(\mathcal{A}(S), c) \right] &= \mathbb{E}_{c,S} \left[ \mathsf{R}_\rho^D(\mathcal{A}(S), c) \mid \mathcal{E} \right] && \text{(By construction of } D) \\
&= \frac{1}{2} \mathbb{E}_S \left[ \mathsf{R}_\rho^D(\mathcal{A}(S), c_1) + \mathsf{R}_\rho^D(\mathcal{A}(S), c_2) \mid \mathcal{E} \right] && \text{(Random choice of } c) \\
&\geq \frac{1}{2} \mathbb{E}_S \left[ \mathsf{R}_\rho^D(c_1, c_2) \mid \mathcal{E} \right] && \text{(Lemma 22)} \\
&= \frac{1}{2} . && \text{(Equation 2)}
\end{aligned}
$$

$\square$

## C.2 Proof of Lemma 3

*Proof.* Fix a target concept $c \in \mathcal{C}$ and the target distribution $D$ over $\mathcal{X}$. Define a hypothesis $h$ to be "bad" if $R_\rho^D(c, h) \geq \epsilon$. Note that any robust ERM algorithm will be robustly consistent on the training sample by the realizability assumption. Let $\mathcal{E}_h$ be the event that $m$ independent examples drawn from $\mathsf{EX}(c, D)$ are all robustly consistent with $h$. Then, if $h$ is bad, we have that $\Pr(\mathcal{E}_h) \leq (1 - \epsilon)^m \leq e^{-\epsilon m}$. Now consider the event $\mathcal{E} = \bigcup_{h \in \mathcal{H}} \mathcal{E}_h$. We have that, by the union bound,

$$\Pr(\mathcal{E}) \leq \sum_{h \in \mathcal{H}} \Pr(\mathcal{E}_h) \leq |\mathcal{H}| \, e^{-\epsilon m} \ .$$

Then, bounding the RHS by $\delta$, we have that whenever $m \geq \frac{1}{\epsilon} \left( \log |\mathcal{H}_n| + \log \frac{1}{\delta} \right)$, no bad hypothesis is *robustly* consistent with $m$ random examples drawn from $\mathsf{EX}(c, D)$. If a hypothesis is not bad, it has robust risk bounded above by $\epsilon$, as required. $\qquad\square$

## C.3 Proof of Lemma 5

*Proof.* The proof is very similar to the VC dimension upper bound in PAC learning. The main distinction is that instead of looking at the error region of the target and any function in $\mathcal{H}$, we must look at its $\rho$-expansion. Namely, we let the target $c \in \mathcal{C}$ be fixed and, for $h \in \mathcal{H}$, we consider the function $(c \oplus h)_\rho : x \mapsto \mathbf{1}[\exists z \in B_\rho(x) \ . \ c(z) \neq h(z)]$ and define a new concept class $\Delta_{c,\rho}(\mathcal{H}) = \{(c \oplus h)_\rho \mid h \in \mathcal{H}\}$. It is easy to show that $\mathsf{VC}(\Delta_{c,\rho}(\mathcal{H})) \leq \mathsf{RVC}_\rho(\mathcal{C}, \mathcal{H})$, as any sign pattern achieved on the LHS can be achieved on the RHS.

The rest of the proof follows from the definition of an $\epsilon$-net and the bound on the growth function of $\Delta_{c,\rho}(\mathcal{H})$, and is included for completeness.

First, define the class $\Delta_{c,\rho,\epsilon}(\mathcal{H})$ as $\left\{ \tilde{c} \in \Delta_{c,\rho}(\mathcal{H}) \mid \Pr_{x \sim D}(\tilde{c}(x) = 1) \geq \epsilon \right\}$, i.e., the set of functions in $\Delta_{c,\rho}(\mathcal{H})$ which have a robust risk greater than $\epsilon$. Recall that a set $S$ is an $\epsilon$-net for $\Delta_{c,\rho}(\mathcal{H})$ if for every $\tilde{c} \in \Delta_{c,\rho,\epsilon}(\mathcal{H})$, there exists $x \in S$ such that $\tilde{c}(x) = 1$. We want to bound the probability that a sample $S \sim D^m$ fails to be an $\epsilon$-net for the class $\Delta_{c,\rho}(\mathcal{H})$, as if $S$ is an $\epsilon$-net, then any robustly consistent $h \in \mathcal{H}$ on $S$ will have robust risk bounded above by $\epsilon$. As with the standard VC dimension, a sample $S$ will be drawn in two phases. First draw a sample $S_1 \sim D^m$ and let $\mathcal{E}_1$ be the event that $S_1$ is not an $\epsilon$-net for $\Delta_{c,\rho}(\mathcal{H})$. Now, suppose $\mathcal{E}_1$ occurs. This means there exists $\tilde{c} \in \Delta_{c,\rho,\epsilon}(\mathcal{H})$ such that $\tilde{c}(x) = 0$ for all the points $x \in S_1$. Fix such a $\tilde{c}$ and draw a second sample $S_2 \sim D^m$. Then, letting $X$ be the random variable representing the number of points in $S_2$ that are such that $\tilde{c}(x) = 1$, we can use Chernoff bound to show that

$$\Pr(X < \epsilon m/2) \leq 2 \exp\left( -\frac{\epsilon m}{12} \right) \ , \tag{3}$$

ensuring that whenever $\epsilon m \geq 24$, the probability that at least $\epsilon m/2$ points in $S_2$ satisfy $\tilde{c}(x) = 1$ is bounded below by $1/2$.

Now, consider the event $\mathcal{E}_2$ where a sample $S = S_1 \cup S_2$ of size $2m$ such that $|S_1| = |S_2| = m$ is drawn from $\mathsf{EX}(c, D)$ and there exists a concept $\tilde{c} \in \Pi_{\Delta_{c,\rho,\epsilon}(\mathcal{H})}(S)$ such that $|\{x \in S \mid \tilde{c}(x) = 1\}| \geq \epsilon m/2$ and $\tilde{c}(x) = 0$ for all $x \in S_1$, where $\Pi_{\Delta_{c,\rho,\epsilon}(\mathcal{H})}(S)$ is the set all possible dichotomies on $S$ induced by $\Delta_{c,\rho,\epsilon}(\mathcal{H})$. Then $\Pr(\mathcal{E}_2) \geq \frac{1}{2} \Pr(\mathcal{E}_1)$ from Equation 3. Now, the probability that $\mathcal{E}_2$ happens for a fixed $\tilde{c} \in \Delta_{c,\rho,\epsilon}(\mathcal{H})$ is

$$\frac{\binom{m}{\epsilon m/2}}{\binom{2m}{\epsilon m/2}} \leq 2^{-\epsilon m/2} \ .$$

Finally, letting $d = \mathsf{RVC}_\rho(\mathcal{C}, \mathcal{H})$ we can bound the probability of $\mathcal{E}_1$ using the union bound:

$$\Pr(\mathcal{E}_1) \leq 2\Pr(\mathcal{E}_2)$$

$$\leq 2 \left| \Pi_{\Delta_{c,\rho,\epsilon}(\mathcal{H})}(S) \right| 2^{-\epsilon m/2}$$

$$\leq 2 \left| \Pi_{\Delta_{c,\rho}(\mathcal{H})}(S) \right| 2^{-\epsilon m/2}$$

$$\leq 2 \left( \frac{2em}{d} \right)^d 2^{-\epsilon m/2} \ . \qquad \text{(Sauer's Lemma)}$$

Thus, there exists a universal constant such that provided $m$ is larger than the bound given in the statement of the theorem, $\Pr(\mathcal{E}_1) < \delta$, as required. $\qquad\square$

### C.4 Proof of Theorem 7

*Proof.* The sample complexity bounds come from Lemmas 3 and 5 and the fact that the Standard Optimal Algorithm (SOA) is a consistent learner, as it will be given counterexamples in the perturbation region until a robust loss of zero is achieved.

For each query to LEQ, a counterexample is returned, or the robust loss is zero. Then, using the mistake upper bound of SOA, which is $\mathsf{Lit}(\mathcal{C})$, we get the query upper bound. $\qquad\square$

### C.5 Proof of Theorem 9

*Proof.* Let $c$ be the target conjunction and let $D$ be an arbitrary distribution. We describe an algorithm $\mathcal{A}$ with polynomial sample and query complexity with access to a $\rho$-LEQ oracle. By Lemma 3, if we can get guarantee that $\mathcal{A}$ returns a hypothesis with zero robust loss on a i.i.d. sample of size $m = O\left(\frac{1}{\epsilon}\left(n + \log\frac{1}{\delta}\right)\right)$ with a polynomial number of queries to the $\rho$-LEQ oracle, we are done.

The algorithm is similar to the standard PAC learning algorithm, in that it only learns from positive examples. Indeed, the original hypothesis $h$ is a conjunction of all of the $2n$ literals. After seeing a positive example $x$, $\mathcal{A}$ removes from $h$ the literals $\bar{x}_i$ for $i = 1, \ldots, n$, as they cannot be in $c$. Note that, by construction, any hypothesis $h$ returned by $\mathcal{A}$ always satisfies $c \subseteq h$[11]. Thus, any counter example returned by the LEQ oracle will have that $c(z) = 1$ and $h(z) = 0$. This allows us to remove at least one literal from the hypothesis set for every counter example. Now, it is easy to see that, for $c \subseteq h' \subseteq h$, if the robust loss $\mathbf{1}[\exists z \in B_\lambda(x) . c(z) \neq h(z)]$ on $x$ w.r.t. $h$ is zero, so will be the robust loss on $x$ w.r.t. the updated hypothesis $h'$. Hence, $\mathcal{A}$ makes at most $m + 2n$ queries to the LEQ oracle. $\qquad\square$

### C.6 Proof of Theorem 10

*Proof.* The sample complexity bound uses Lemma 3. Note the class $\mathsf{LTF}^W_{\{0,1\}^n}$ has size $O(2^n(n + W)^{\min\{n,W\}})$. This is a simple application of the stars and bars identity, where $W$ is the number of stars and $n + 1$ the number of bars (as we are considering the bias term as well): $\binom{n+W}{W} = O((n+W)^{min\{n,W\}})$. The $2^n$ term comes from the fact that each weight can be positive or negative. The query complexity uses the fact that the mistake bound for Winnow for $\mathsf{LTF}^W_{\{0,1\}^n}$ is $O(W^2 \log(n))$ in the case of positive weights (the full statement can be found in Appendix B). Littlestone (1988) outlines how to use the Winnow algorithm when the linear classifier's weights can vary in sign, at the cost of doubling the input dimension and weight bound (see Theorem 10 and Example 6 therein). $\qquad\square$

### C.7 Proof of Theorem 11

The proof of this theorem mainly relies on deriving an upper bound on the robust VC dimension of halfspaces. This will help us bound the sample complexity needed to guarantee robust accuracy. The query complexity upper bound follows from this upper bound and the mistake bound for the Perceptron algorithm. To bound the robust VC dimension of linear classifiers, we will need the following theorem from Goldberg and Jerrum (1995):

**Theorem 28** (Theorem 2.2 in Goldberg and Jerrum (1995))**.** *Let $\{\mathcal{C}_{k,n}\}_{k,n\in\mathbb{N}}$ be a family of concept classes where concepts in $\mathcal{C}_{k,n}$ and instances are represented by $k$ and $n$ real values, respectively. Suppose that the membership test for any instance $\alpha$ in any concept $C$ of $\mathcal{C}_{k,n}$ can be expressed as a boolean formula $\Phi_{k,n}$ containing $s = s(k,n)$ distinct atomic predicates, each predicate being a polynomial inequality or equality over $k + n$ variables (representing $C$ and $\alpha$) of degree at most $d = d(k,n)$. Then $\mathsf{VC}(\mathcal{C}_{k,n}) \leq 2k\log(8eds)$.*

---

[11]We overload $c, h$ to mean both the functions and the set of literals in the conjunction, as it will be unambiguous to distinguish them from context.

We will now translate the $\rho$-expansion of the error region (i.e., the robust loss function) between two halfspaces as a boolean formula using a result from Renegar (1992). This will allow us to use the theorem above from Goldberg and Jerrum (1995) to bound the robust VC dimension of $\mathsf{LTF}_{\mathbb{R}^n}$.

**Lemma 29.** *Let $a, b \in \mathbb{R}^n, a_0, b_0 \in \mathbb{R}$, and define the map $\varphi : x \mapsto \mathbf{1}[\exists z \in B_\rho(x) \ . \ sgn(a^\top z + a_0) \neq sgn(b^\top z + b_0)]$. Then $\varphi$ can be represented as a boolean formula $\Phi$ with $s = 10^{Cn^2}$ distinct atomic predicates, each predicate being a polynomial inequality over $2n + 2$ variables of degree at most $10^{C'n}$ for some constants $C, C' > 0$.*

*Proof.* First note that the predicate $sgn(a\top z + a_0) \neq sgn(b^\top z + b_0)$ can be represented as the following formula:

$$\left(a^\top z + a_0 \geq 0 \wedge b^\top z + b_0 < 0\right) \vee \left(a^\top z + a_0 < 0 \wedge b^\top z + b_0 \geq 0\right) \ ,$$

which contains $n + (2n + 2)$ variables and 4 predicates. Moreover, given a perturbation $\zeta \in \mathbb{R}^n$, the constraint $\|\zeta\|_2 \leq \rho$ on its magnitude is a polynomial inequality of degree 2:

$$\sum_i \zeta_i^2 \leq \rho^2 \ .$$

Now, consider the following formula:

$$\Psi(x) = \exists \zeta \in \mathbb{R}^n \ . \ \left(sgn(a\top z + a_0) \neq sgn(b^\top z + b_0) \wedge \|\zeta\|_2 \leq \rho\right) \ .$$

This is a formula of first-order logic over the reals. Using the notation of Theorem 27, we have $\omega = 1$ quantifier, and thus $\prod_k n_k = n$, one Boolean formula with $m = 5$ polynomial inequalities of degree $d$ at most 2, and $l = n$. Thus, $\Psi(x)$ can be expressed as a quantifier-free formula $\Phi(x) = \bigvee_{i=1}^I \bigwedge_{j=1}^{J_i} (h_{ij}(y) \Delta_{ij} 0)$ of size

$$I \max_i J_i \leq (md)^{2^{O(\omega)} l \prod_k n_k + 2^{O(\omega)} \prod_k n_k} \leq 10^{Cn^2}$$

for some constant $C$, where the polynomial inequalities are of degree at most $(md)^{2^{O(\omega)} \prod_k n_k} \leq 10^{C'n}$ for some constant $C'$. $\qquad\square$

We thus get the following corollary.

**Corollary 30.** *The robust VC dimension of $\mathsf{LTF}_{\mathbb{R}^n}$ is $O(n^3)$.*

*Proof.* We let $s = 10^{Cn^2}$, $k = 2n + 2$ and $d = 10^{C'n}$ from the proof above and use Definition 4 and Theorem 28 to get a robust VC dimension upper bound of $O(k \log(sd)) = O(n^3)$.[12] $\qquad\square$

Proving Theorem 11 is now a straightforward application of the results above.

*Proof of Theorem 11.* The sample complexity upper bound is a consequence of Corollary 30 and Lemma 5. The query complexity upper bound follows from Lemma 8 and the mistake bound for the Perceptron algorithm, which appears in Appendix B. $\qquad\square$

---

[12]Note that Corollary 2.4 in Goldberg and Jerrum (1995) uses this reasoning.