# OpenReview forum: "When are Local Queries Useful for Robust Learning?"
_NeurIPS.cc/2022/Conference — NeurIPS 2022 Accept_

### Official Review · Reviewer_5J5m · 2022-06-23

**Rating:** 6
**Confidence:** 3
**Soundness:** 3 good
**Presentation:** 3 good
**Contribution:** 3 good

**Summary:**

The paper studies adversarial robustness of learning basic concepts when the learner is allowed to make local queries. It shows that

- Using local membership queries of Awasthi et al (2013) does not help too much on reducing the robust risk. This is in stark contrast to results established under the standard PAC model.

- It then shows that with access to a new query type, equivalence query, the robust risk can be reduced.

In terms of merit, this paper gives the first robust ERM algorithm when robust risk is defined as being exact-in-the-ball.

In terms of techniques, this paper leverages many useful prior results and seems to combine them nicely.

**Questions:**

Please address the weakness part above.

**Ethics Review Area:**

["I don’t know"]

**Limitations:**

Yes.

**Strengths And Weaknesses:**

Strength:

+ The paper studies a timely topic and gives full proof to its main results.
+ It is interesting to consider various oracles to tackle the essentially hard problem of learning with adversarial examples. In particular, the separation result for local membership query appears interesting.

Weakness:

- It turns out that the main positive result is the improvement from using equivalence queries. However, I feel such query type was not well motivated, and seems too strong to find practical applications.

- There are prior works studying the problem through the lens of online learning, which does not rely on the strong oracles, see e.g.

The Complexity of Adversarially Robust Proper Learning of Halfspaces with Agnostic Noise, I. Diakonikolas, D. Kane, P. Manurangsi, NeurIPS 2020

The above paper already sets out PAC learnability of large-margin halfspaces via L_p-norm Perceptron. I would like to see a close comparison to Theorem 11 in the submission.

- While the paper presents a handful set of theoretical results, a highlight of the core techniques is missing. Thus, it is unclear to me whether the developed techniques are novel or the established results are just simple sandwich of prior works.

- It turns out that the main claim is a robust ERM when adversarial examples reside exact-in-the-ball while prior works have built upon the regime constant-in-the-ball. I wonder what is the key challenge in the current setting and why existing results cannot be easily adapted.

---

> ### Author Response · Authors · 2022-07-30
> **Reply to Reviewer 5J5m**
>
> Dear reviewer,
>
> We thank you very much for your comments and questions, which are addressed below.
>
> **Local equivalence queries**:
>
> - Regarding applications, if one considers the exact-in-the-ball notion of robustness, the existence of adversarial examples (and thus an entity that can find them) can be seen as a justification for an LEQ oracle. Indeed, if an algorithm can find a point in the perturbation region where the ground truth and hypothesis disagree (with a human in the loop to confirm this), then this can be used as a counterexample to the learning algorithm. While this may be harder to implement in practice, it is in line with active learning approaches. We will include a discussion on this in the revised version.
> - Regardless of practicalities, oracles such as the LEQ have been studied in theory and have resulted in valuable insights. For example, the EQ model (where the queries are made on the whole input space) has been studied in learning theory and automata theory. Our model is a restricted version of this.
> - Since our work wishes to answer, among other questions, when robust learning is possible, our negative results for LMQ and $\lambda$-LEQ where $\lambda<\rho$ implicitly justify the use of the $\rho$-LEQ oracle.
>
> **Diakonikolas et al 2020**:
>
> This paper considers the constant-in-the-ball notion of robustness (last paragraph on p.1). While both papers use the same algorithm, the guarantees are obtained in a different way, and have a different meaning (as the robust risk minimizer won’t necessarily be the same for the same learning problem with two different notions of robustness). [DKM20] also study the agnostic setting, while we study the realizable one (though both use a different notion of risk, so a problem that could be agnostic w.r.t. their notion of risk could be realizable for us). We will make sure to add a reference and a brief discussion.
>
> **Robust ERM**:
>
> While showing that RERM results in robust learners is not fundamentally different from similar results in the constant-in-the-ball (or even standard PAC) setting, the main difficulty is either designing RERM algorithms for exact-in-the ball risk (or showing that existing algorithms already are). For example, the notion of Robust VC dimension is also different in the two settings. Theorems 10 and 11 establish the existence of robust learning algorithms. We emphasize that designing RERM algorithms for this notion of risk (even without computational constraints) is hard/impossible without access to some oracle like the LEQ one. For the constant-in-the-ball notion of robustness, once we have a sufficiently large sample to guarantee robust generalization, the issue with computing the robust risk minimizer is mainly a _computational_ one, as we wish to know when the hypothesis changes label in the perturbation region. Montasser et al. (2021) introduced the Perfect Adversary Oracle for the cases where this computation is inefficient or impossible. On the other hand, the LEQ tackles both the _information-theoretic_ and computational limitations of learning with only access to a random sample.
> Highlight of the core techniques: page limit allowing, we would like to include some proofs from the appendix (or sketches/more thorough explanations). We would like to point towards the response to reviewer n77F regarding the originality/technicality of the proofs.
>
> Thank you for your time, and we are looking forward to your response.

---

> > ### Comment · Reviewer_5J5m · 2022-08-09
> > **reviewer response**
> >
> > Thank you for the reply. I guess my concerns were not addressed.
> >
> > - On the LEQ: as far as I can tell, EQ is not broadly studied in the literature. So it is still unclear to why when LEQ will be useful. (I know this is a theory paper, but NeurIPS is a machine learning conference rather than pure math venue; people care about when and how one can apply the theory.)
> >
> >
> > - "While this may be harder to implement in practice, it is in line with active learning approaches." Can you explain more on the connection to active learning?
> >
> > - [Dia20]: "While both papers use the same algorithm, the guarantees are obtained in a different way, and have a different meaning." Can you explain more on "different meaning"?
> >
> > - constant-in-the-ball vs exact-in-the-ball: this is the main claim of the paper, but to be very honest, even after reading the rebuttal and the submitted manuscript, it is unclear to me why there is a significant gap between the two setting and why prior works on one cannot be adapted/generalized to the other.

---

> > > ### Author Response · Authors · 2022-08-09
> > > **Response #1 to reviewer's concerns**
> > >
> > > Thank you for taking the time to read our rebuttal, and we hope that our response below addresses your remaining concerns. We have split our response into two parts due to character constraints.
> > >
> > > **The constant-in-the-ball** results cannot, in general, be applied to the exact-in-the-ball setting.
> > > - In terms of techniques, the guarantees obtained by Montasser et al (2021) (we are referring to this version for theorem numbers http://proceedings.mlr.press/v134/montasser21a/montasser21a.pdf) are closest in essence to our work. These guarantees, which notably give a query upper bound as a function of the VC and dual VC dimensions of a concept class (Theorem 2), cannot be applied to the exact-in-the-ball notion of robustness. This is because their argument (which is based on earlier results, Montasser et al (2019)) relies on inflating the training sample $S$ as follows: for $(x,y)\in S$ and perturbation function $\mathcal{U}$, add all the points in $\mathcal{U}(x)$ to $S$ *with label $y$* (this is not what we want for our notion of robustness: we would want the ground truth). Of course, this set can now be infinite. The authors discretize the set to a new set of cardinality $\tilde{O}(VC(H)VC^*(H))$, where $VC^*(H)$ is the dual VC dimension. Their argument is based on assuming that the ground truth is constant in the perturbation region, as they are looking at the dual set of function $\mathcal{G}$, defined as $g(x,y) (h) = 1[h(x) \neq y] $ (see p.7 of the following version: http://proceedings.mlr.press/v99/montasser19a/montasser19a.pdf). This is not possible to do with the exact-in-the-ball notion of robustness. This reasoning is based on the constant-in-the-ball realizable setting, and the agnostic setting is based on a reduction to the realizable setting. As previously mentioned, the constant-in-the-ball agnostic setting could be the exact-in-the-ball realizable, so this reduction does not work. Of course, when the notions of robustness coincide (there is a sufficiently large margin, for example), we could use their reasoning, but it is not true in general, for instance when there is considerable probability mass near the boundary. Finally, for many finite cases, our bounds are significantly smaller than those of Theorem 5 in Montasser 2021 ($VC(H)Lit(C)$ vs $2^{VC(H)^2{VC^*}^2(H)}Lit(C)$), simply due to the nature of the problem studied. Both their and our work use online learning guarantees to derive the $Lit(C)$ part of the upper bound.
> > > - Another important part of the gap is that, for the same learning problem, the two notions of robustness can imply different solutions (robust risk minimizers), and thus have different implications. For a more applied setting, if we consider the example we gave to reviewer n77F, when we have medical data, changes to a small number of observations may change the diagnosis; here, we would _not_ want the classifier returned by a learning algorithm to be constant-in-the-ball. For a more theoretical example, if we consider the class of parities (functions that are expressed as $f(x)=\sum_{i\in S}x_i \mod 2$, for some subset $S\subseteq[n]$), then all points are on the decision boundary. Under the uniform distribution, the exact-in-the-ball risk minimizer is the target $f$, but for the constant-in-the-ball, it is the constant function 0 (or 1). This is also an example of a problem that is realizable for the exact-in-the-ball robust risk and agnostic for the constant-in-the-ball one.
> > >
> > > Please let us know if it would be useful to include a discussion summarizing the points above.

---

> > > ### Author Response · Authors · 2022-08-09
> > > **Response #2**
> > >
> > > We reply to the remaining concerns below. Please let us know if you have any questions.
> > >
> > > **For the LEQ**, if one believes that the exact-in-the-ball notion of robust risk is worth investigating, then our lower bound for the LMQ model and our impossibility result for $\lambda<\rho$-LEQ give ample justification for looking at the $\rho$-LEQ to delimitate when distribution-free robust learning is possible in this setting. Using the perfect adversary oracle (PAO) of Montasser et al (2021) could give false counterexamples, for example a point $z$ where $h(z)\neq c(x)$, but where $h(z)=c(z)$ (i.e. the hypothesis is correct on $z$ w.r.t. the ground truth). The LEQ setting is of course idealized, and it would be worth looking at imperfect oracles in future work. We would also like to note that NeurIPS has a longstanding learning theory branch, which routinely features papers that don’t have immediate applications, but provide substantial insights to various learning problems.
> > >
> > > **The EQ+MQ model:** this learning model has been vastly studied since the seminal work of Angluin (1987), cited by over 2600, as outlined in the related work. More recently in the machine learning community, EQs (and MQs) were used for recurrent neural networks in [WGY18] and [WGY19]. [O+20] later followed up on their work by showing how regression-based weighted finite automata extraction could be used to simulate EQs, both theoretically and experimentally. [SDC19] also used the MQ+EQ model to verify binarized neural networks, where the EQs were simulated by a SAT solver. [CM19] also uses MQ+EQ for interpretable models based on linear temporal logic. We are happy to include these references in the related work section.
> > >
> > > **[Dia20]**: since we are using different notions of robustness, the hypotheses returned by both algorithms may differ (recall that our algorithm is given counterexamples that the one from [Dia20] will not have access to). Guarantees derived for the constant-in-the-ball notion of robustness will imply that the hypothesis returned has a certain *stability* (perhaps at the cost of accuracy in the agnostic case), because we are trying to limit the probability of a label change in the perturbation region. On the other hand, guarantees derived for the exact-in-the-ball notion of robustness usually give stronger accuracy, as we want to be correct wrt the ground truth in the perturbation region. Apologies for the ambiguous wording.
> > >
> > > **Active learning**: in active learning, an algorithm can interact with a (human) user or information source (which can be represented as an oracle). This is usually done by querying the label of a data point. By having a human user in the loop, we can ensure that the counterexamples given to the learner are indeed counterexamples w.r.t. the exact-in-the-ball robust risk (which is necessary for the guarantees we have proven to hold). Here the human is required as we cannot assume that the label is constant in the perturbation region, contrary to existing methods to find adversarial examples in the literature. We will include a short discussion on active learning and how it could relate to the LEQ in practice.
> > >
> > > Thank you for pointing out your concerns and engaging with us during the discussion period! It has been beneficial to clarify and justify our work. We are looking forward to your response.
> > >
> > > **References**
> > >
> > > [WGY18] Gail Weiss, Yoav Goldberg, and Eran Yahav. *Extracting Automata from Recurrent Neural Networks Using Queries and Counterexamples*, ICML 2018.
> > >
> > > [WGY19] Gail Weiss, Yoav Goldberg, and Eran Yahav. *Learning Deterministic Weighted Automata with Queries and Counterexamples*, NeurIPS 2019.
> > >
> > > [O+20] Takamasa Okudono, Masaki Waga, Taro Sekiyama, and Ichiro Hasuo. *Weighted Automata Extraction from Recurrent Neural Networks via Regression on State Spaces*, AAAI 2020.
> > >
> > > [SDC19] Andy Shih, Adnan Darwiche, and Arthur Choi. *Verifying binarized neural networks by Angluin-style learning*, International Conference on Theory and Applications of Satisfiability Testing  2019.
> > >
> > > [CM19] Alberto Camacho, Sheila A. McIlraith. *Learning Interpretable Models Expressed in Linear Temporal Logic*, International Conference on Automated Planning and Scheduling (ICAPS 2019).

---

### Official Review · Reviewer_u9Eu · 2022-07-11

**Rating:** 6
**Confidence:** 3
**Soundness:** 3 good
**Presentation:** 3 good
**Contribution:** 2 fair

**Summary:**

The authors study adversarially robust learning in the "exact-in-a-ball" model when the learner has access to additional types of local queries. The `exact-in-a-ball' robustness model replaces the standard PAC classification error with

$$\Pr_{x \sim D}[\exists z \in B(x,r): h(z) \neq c(z)]$$

This differs from the widely studied "constant-in-a-ball" model that instead uses $h(x) \neq c(z)$. Since learning even basic concepts in this model is typically impossible, the authors examine two types of enriched query algorithms: local membership queries and local equivalence queries. The former allows the learner to query any point in a small ball around any training example, and the latter allows the learner to ask whether a given hypothesis h is robust on a small ball around the point.

The authors show that while local membership queries do not help robust learning even in basic cases such as conjuctions over the uniform distribution, local equivalence queries can be used to robustly learn several non-trivial hypothesis classes such as conjuctions and halfspaces with margin, even in the distribution-free setting. In fact, the authors more generally show it is possible to robustly learn any finite class or class with finite robust VC dimension that has an online learner in the mistake bound model. Their algorithm is computationally efficient as long as the online learner is as well, which leads to statistically and computationally efficient robust learners for conjunctions and halfspaces with margin.


**Questions:**

A few typos: attaks, dimensino, ispoerimetric

**Strengths And Weaknesses:**

The `exact-in-a-ball' model of adversarial robustness is a challenging learning model, and this work gives the first efficient robust learners for important classes such as halfspaces with margin. While most of the general techniques in the work are not particularly novel (e.g. Robust VC, use of online learning), the eventual application and analysis showing halfspaces actually satisfy these conditions (namely Robust VC) is nice. Overall, the results are a bit niche, but may be of interest to the adversarial learning sub-community.

On the other hand, the paper can be a bit confusing at places, and also seems to have some small errors. Most of the sample complexities for instance seem to be missing a $\log(1/\varepsilon)$ term that should appear from applying Sauer-Shelah. The statement of results in Section 3.2 are also a bit confusing and stated in a non-standard fashion (perhaps a more typical presentation would be to say any RERM on m samples is a robust learner or something of this sort). Finally, while the authors largely do a good job placing the result in prior literature, it should probably also be mentioned that online learning (namely perceptron) has also been used in other work on robust learning, e.g. in [BJC21] “Sample Complexity of Robust Linear Classification on Separated Data.”

---

> ### Author Response · Authors · 2022-07-30
> **Reply to Reviewer u9Eu**
>
> Dear reviewer,
>
> We thank you very much for your comments. Regarding your questions:
>
> $\log(1/\epsilon)$ term: You are absolutely right, the $\log(1/\epsilon)$ term was omitted by mistake in the statement of Lemma 5. It is not necessary for Lemma 3 however. Thank you for pointing this out.
>
> Statements in Section 3.2: We agree and we will rephrase the statements in line with your comments.
>
> [BJC21] reference: thank you for pointing this out to us, we will make sure to include it in the revision. Lines 329-338 include a short discussion on when our results differ from when considering the constant-in-the-ball loss, which covers the [BJC21] paper.
>
> Finally, thank you for pointing out the typos!

---

### Official Review · Reviewer_n77F · 2022-07-12

**Rating:** 6
**Confidence:** 1
**Soundness:** 4 excellent
**Presentation:** 4 excellent
**Contribution:** 2 fair

**Summary:**

The paper considers (adversarial) robust learning in the PAC learning framework. As opposite to the "standard" notion of adversarial robustness, where the hypothesis $h$ is considered to be robust at point $x$ with radius $\epsilon$ if the target concept is $c$ and it holds that $||x'-x|| \leq \epsilon \implies h(x')= c(x)$. Here, the hypothesis needs to be correct in the epsilon ball to be considered robust. That is, it needs to satisfy $||x'-x|| \leq \epsilon \implies h(x')= c(x')$. To achieve this, the learner is allowed to use "Local membership queries" in at-most $\lambda$ distance from the observed point. The paper then shows that when $\lambda < \epsilon$, there is a distribution not learnable by any learner (thm 2). Then they provide several robust learnability results (thms 7,9,10,11) on classes such as linear threshold functions and conjunctions.

**Questions:**

- Please define the distribution-free setting. Does it mean that there are no assumptions on the distribution?  The term suggests that there is no underlying distribution, but then the notion of random sample would be meaningless.  Also I think it would be good if all definitions were in the appendix. Now the majority is there, but consistency is in the main text.

- Why is correct-in-the-ball notion of robustness interesting? Specially, in this setting, we would like to be correct at every point (which is the same goal as we usually have, as opposite to the constant-in-the-ball risk), thus, the learning is basically the same as in the standard setting. I think there should be at least a brief discussion in the paper about this.

- Feel free to point out any potential errors in the review.

## misc
- footnote 4 - why is Hamming distance the only meaningful distance here? While I agree that is a reasonable distance, I think there are many others and the space can have a special structure which can be of interest. E.g., If there is an underlying graph structure (other than a hypercube) which would induce the distances so that the "adversarial neighborhood" may contain e.g., neighboring nodes.





**Limitations:**

-

**Strengths And Weaknesses:**



# originality, clarity, quality, significance
- Originality - The proof techniques seem to be standard (often the proofs are trivial modification of analogical proofs in the standard setting) but the considered problems are original, so I think the originality is ok. However, if there is some original technique that I have missed, I suggest the authors to emphasize it.
- Clarity - Clearly written. Only that there are used lots of abbreviations which slows down the reading, but their presence is understandable.
- Quality - I haven't found a problem.
- Significance - I don't consider this work to be "too" significant, especially because the results are often unsurprising and easy to derive, but I think it has its place.


# Pros
- Easy to read
- Extensive amount of previous work in the appendix, although something is superfluous.
- Quite a lot of results.

# Cons
- The results are sometimes too simple, such as thm 2. which is claimed as one of the main contributions essentially says that when we cannot observe c(x) during training, then we don't know it. For lemma 3, I don't see a difference between the proof and a proof of the lemma in the standard (non-robust) setting (and then thm 9 is a corollary of lemma 3), Similarly lemma 5 seems to only replace VC dimension with robsut VC dimension, but there is almos no discussion on RVC (e.g., what is the RVC of standrd function classes) and so on.

---

> ### Author Response · Authors · 2022-07-30
> **Reviewer n77F reply**
>
> Dear reviewer,
>
> We thank you very much for your comments and questions, which are addressed below.
>
> **Originality**:
>
> For the proofs that use standard methods, we believe that the conceptual contribution (i.e., robust ERM guarantees in the distribution-free setting) is important, as it contrasts with the natural setting of having only random examples, where this is impossible. Moreover, Theorem 7 uses online learning guarantees but also a bound on the size of the Littlestone tree, giving an upper bound with a dependence on the adversarial budget $\rho$, which doesn’t appear in related work for the constant-in-the-ball notion of robustness (see lines 151-153 in the related work section).
>
> There are more technical proofs in the paper, which had to be relegated to the appendix because of space constraints, particularly Theorems 11 and 12. Theorem 11 transforms the robust risk function into a first-order logic formula and uses the quantifier-elimination procedure of Renegar (1992) to bound the robust VC dimension of halfspaces with margin. Theorem 12 provides an LMQ lower bound (which interestingly also holds for full membership queries, where queries can be made over the whole input space). We would be happy to include either or both of the proofs in the main body of the paper, page limit permitting, and at the very least include a summary of the technical contributions in the main body of the paper.
>
> **Cons**:
>
> Our results in the rest of Section 3 derive bounds using an LEQ oracle with $\lambda = \rho$. In order to justify that, we do feel the need to include Theorem 2 (although the proof is very simple) for conceptual completeness – it is stated as a contribution for that reason.
>
> Lemmas 3 & 5: We mention in the submission that the arguments are standard. However, the versions of these for the exact-in-the-ball setting don’t appear elsewhere and need to be included for completeness. Regarding Theorem 9 – it is not true that it is a direct corollary of Lemma 3; the analysis of consistency is different and is better than can be directly derived from the more general result in Lemma 8.
>
> Regarding discussion on Robust VC dimension: this does appear implicitly in Theorem 11, where the RVC of linear thresholds is shown to be $O(n^3)$. We agree that more discussion would be useful, and space permitting we will add this. For other classes we considered in the paper, this discussion was not needed, as they are finite and Lemma 3 already applies.
>
> **Questions**:
>
> The distribution-free setting refers to settings where learning guarantees hold regardless of the distribution that generated the data (as long as the examples are sampled iid). This is in fact the setting for PAC Learning (Definition 14 in the appendix illustrates this). Related work has shown that the distribution-free setting is impossible for robust learning when having access to only random samples. We show that the impossibility result carries over (though to more restricted concept classes) even when giving a lot more power to the learner.
>
> Exact-in-the-ball-robustness: robust (exact-in-the-ball) learning is not the same as in the standard learning, because the notion of robustness (l.165) contains an existential quantifier: we want to be correct in the perturbation region (which may contain points that are less likely to be seen in a random sample), not just on the random sample. It is a much stronger requirement than that of standard learning.
>
> The exact-in-the-ball notion of robustness is much less studied and understood than the constant-in-the-ball one, which is one of the reasons why we focus on it. We also highlight that, when considering problems other than image classification with small perturbations, there are many examples where the notion that we are using is more justified. This usually happens when there is considerable probability mass near the decision boundary (and thus, a label change in the hypothesis could represent a ground truth label change!). For example, when we have medical data, changes to a small number of observations may change the diagnosis; here, we would _not_ want the classifier to be constant-in-the-ball.
>
> Hamming distance: the Hamming distance is the only meaningful notion of distance in the boolean hypercube, but, as you mentioned, other input spaces like a graph or $\mathbb{R}^n$ have various meaningful notions of distance. We will remove the footnote.
>
> We hope we have answered your concerns and look forward to reading your response.

---

> > ### Comment · Reviewer_n77F · 2022-08-03
> > **Reponse**
> >
> > Thank you for your response. I think that we "agree" on the fact that my review is somewhat accurate. While I might have sounded a bit too critical, I mainly tried to write the review so that no confusion can occur. Overall I quite liked the paper and the reason for only borderline accept is my lack of expertise in this topic; I wanted to not outweight the other (expert) reviews.

---

> > > ### Comment · Reviewer_n77F · 2022-08-08
> > > **change of score**
> > >
> > > I decided to increase the score and decrease the confidence. It might reflect my opinion better.

---

### Public Comment · Authors · 2023-07-21
**Erratum**

Since the final submission, we have discovered an error in Theorem 11. We have published a new version of our work on arXiv (https://arxiv.org/abs/2210.06089), which contains an erratum (Appendix C), as well as a fix resulting in new contributions to the work (Section 3.6).

---

### Meta-Review · Area_Chair_gNXA · 2022-08-24

**Recommendation:** Accept
**Confidence:** Certain

**Metareview:**

Solid contribution to NTK theory

**Award:**

No

---

### Decision · Program_Chairs · 2022-09-14

Accept